# Single-Step Fabrication of Au-Fe-BaTiO_3_ Nanocomposite Thin Films Embedded with Non-Equilibrium Au-Fe Alloyed Nanostructures

**DOI:** 10.3390/nano12193460

**Published:** 2022-10-03

**Authors:** Bethany X. Rutherford, Hongyi Dou, Bruce Zhang, Zihao He, James P. Barnard, Robynne L. Paldi, Haiyan Wang

**Affiliations:** 1School of Materials Engineering, Purdue University, West Lafayette, IN 47907, USA; 2School of Electrical and Computer Engineering, Purdue University, West Lafayette, IN 47907, USA

**Keywords:** Au-Fe alloy, nanoparticles, thin film, nanocomposite, non-equilibrium

## Abstract

Nanocomposite thin film materials present great opportunities in coupling materials and functionalities in unique nanostructures including nanoparticles-in-matrix, vertically aligned nanocomposites (VANs), and nanolayers. Interestingly the nanocomposites processed through a non-equilibrium processing method, e.g., pulsed laser deposition (PLD), often possess unique metastable phases and microstructures that could not achieve using equilibrium techniques, and thus lead to novel physical properties. In this work, a unique three-phase system composed of BaTiO_3_ (BTO), with two immiscible metals, Au and Fe, is demonstrated. By adjusting the deposition laser frequency from 2 Hz to 10 Hz, the phase and morphology of Au and Fe nanoparticles in BTO matrix vary from separated Au and Fe nanoparticles to well-mixed Au-Fe alloy pillars. This is attributed to the non-equilibrium process of PLD and the limited diffusion under high laser frequency (e.g., 10 Hz). The magnetic and optical properties are effectively tuned based on the morphology variation. This work demonstrates the stabilization of non-equilibrium alloy structures in the VAN form and allows for the exploration of new non-equilibrium materials systems and their properties that could not be easily achieved through traditional equilibrium methods.

## 1. Introduction

As the drive for performance and complexity of electronic and optical devices increases, new materials with novel functionalities and multifunctionalities are highly desired [1,2]. Nanocomposite thin film materials and nanoparticles are considered some of the most promising options given their advantages in both size and physical property tunability when compared to traditional single phase films [3,4,5,6,7,8,9,10]. Nanocomposite thin film materials consist of three main different nanostructure types: multilayers, nanoparticles-in-matrix, and vertically aligned nanocomposites (VANs) [3,9]. Each nanocomposite structure has advantages for materials coupling. For example, the nanoparticles-in-matrix structure has very large surface areas that can be functionalized [11,12,13,14]. Multilayer structures have been demonstrated to be used in optical applications, mainly as a meta-material [15]. VANs, a newly developed nanocomposite structure, are of particular interest due to their ability to grow ordered, controlled structures in a single step fashion. VANs present vertical coupling of two phases in terms of size and distribution of nanopillar features. These ordered VAN structures have demonstrated their ability to tune magnetic, superconducting, electrical, and optical properties [16,17].

Beyond the most demonstrated oxide-oxide VANs, many new VAN systems combine metals with oxide materials in systems such as BaTiO_3_ (BTO)-Au, BTO-Fe, BTO-Co, BaZrO_3_-Co, La_0.5_Sr_0.5_FeO_3_ (LSFO)-Fe, ZrO_2_-Co, ZnO-Co, and ZnO-Au [18,19,20,21,22,23]. By including metals instead of oxides, these new oxide-metal VANs have shown to include enhanced and/or new physical properties, especially at room temperature, such as magnetic anisotropy, optical anisotropy, magneto-optical coupling, enhanced second harmonic generation (i.e., non-linearity), and plasmonic resonance [7,8,16,17,24,25]. It is interesting to note that most of the oxide-metal VAN demonstrations have been focused on single metal ones and the demonstration on multi-metal and metal alloy ones are limited [20,24,26]. This is partly due to the difficulty of high-quality co-growth of oxides and multi-metal systems. These alloyed metal nanostructures present enormous opportunities in achieving phase-stable and broad plasmonic resonance properties. For example, in the case of Ag-Au alloyed pillars in a ZnO matrix, a tunable hyperbolic structure in multi-metal alloyed nanopillars is demonstrated [27]. These metallic phases are miscible and thus form well-mixed nanopillars during the deposition process. Effectively, three-phases were used to deposit a resultant two-phase nanocomposite oxide-metal system grown with a single-step method that also demonstrated enhanced physical properties with even more ability to tune the properties.

In this work, the three-phase nanocomposite system of Au-Fe-BTO is explored as a nanocomposite system to evaluate the miscibility of Fe and Au during the pillar formation process in a BTO matrix through a non-equilibrium processing method, i.e., pulsed laser deposition (PLD) in this case. Three-phase VAN systems have shown limited success in demonstrations of Au-BTO-ZnO, Au-Fe-LSFO, TiN-Au-NiO, and BTO-CeO_2_-Co considering the complex three-phase co-growth process; however, three-phase designs have offered other complexities in nanostructure and property designs [7,8,24,25]. Here, BTO is selected for being a ferroelectric perovskite with dielectric and ferromagnetic properties. The lattice parameter of BTO (a_BTO_ = 3.99 Å) is similar to that of SrTiO_3_ (STO) (a_STO_ = 3.91 Å) substrates, making BTO growth on STO easier as it grows well epitaxially; this allows BTO to become the matrix material [8,18,28,29,30,31]. Fe and Au were selected in order to combine plasmonic Au and ferromagnetic Fe in the designs [11,13,24,32]. Au is also one of the least reactive metals and is diamagnetic [24]. Bimetallic nanoparticles of alloyed Au-Fe have attracted interest due to their ability to combine plasmonic and magnetic properties as well as being biocompatible [11,12,13,32]. Under equilibrium conditions, Au and Fe are immiscible and do not alloy under any temperature, instead forming solid solutions when formed together [13,32,33]. This interaction follows the Hume–Rothery rules of alloying as Au has a face-centered cubic (FCC) structure and *α*-Fe grown at temperatures below 868 ℃ has a body-centered cubic (BCC) structure [33,34]. It is interesting to note that under non-equilibrium conditions, such as pulsed laser melting and PLD, alloying is possible [13,32]. At larger ratios of Au:Fe, Au-Fe nanoparticles became homogeneous instead of phase-separated [13,32]. PLD is a great technique for creating epitaxial thin films that stoichiometrically reflect the composition of the target material and quick fabrication and development of new thin film materials. PLD, however, is limited in terms of large-scale production as the laser plume does not distribute material evenly across large surfaces. This problem can be mitigated through substrate rotation or reel-to-reel process during the deposition process [35].

To achieve three-phase nanocomposite thin films with tunable structure and materials properties, the deposition rate will be used to manipulate the growth kinetics of the thin film. Specifically, lower laser frequencies result in longer diffusion time between the laser pulses and thus produce larger pillar dimensions that are fewer in number across the substrate surface [3]. Figure 1 depicts the expected nanostructure based on previous co-grown three-phase nanocomposite studies, where core–shell or partial core–shell structures were achieved with no intermixing or alloying between phases. Microstructure analysis and property measurements including ferromagnetic property measurements, optical transmittance, and ellipsometry measurements were conducted to correlate the laser frequency tuning with the structural and property tuning in the three-phase VANs.

## 2. Results and Discussion

To reveal the microstructure tuning in the Au-Fe-BTO nanocomposite thin film samples deposited under different laser frequencies, transmission electron microscopy (TEM) and scanning transmission electron microscopy (STEM) analysis along with energy dispersive X-ray (EDX) analysis were implemented for all the samples. For the 2 Hz sample, as shown in STEM images and EDX mapping results in Figure 2, there is significant phase separation of the Au and Fe metallic phases in the BTO matrix as the slow deposition rate allowed the Au to diffuse towards the top of the thin film, whereas the Fe diffused towards the bottom of the film (Figure 2A–D). This large separation likely occurs due to the deposition setup of the BTO-Fe target with the Au strip on top; the intermixed nature of the BTO-Fe target allows for the Fe to be deposited towards the bottom of the film and the Au accumulates at the top, mimicking the nature of the target material. In previous work for binary systems, the phase with the lowest surface energy has been shown to self-segregate towards the top of the film [36]. The Au nanoparticles are an average of 9.1 nm in diameter, and the Fe nanoparticles are an average of 8.0 nm in diameter. Figure 2F reflects the nanostructure of the Au-Fe-BTO thin film grown at 2 Hz as a 3D schematic. The plan-view EDX (Figure 2H–K) shows that the density of the Fe nanoparticles across the film is higher than the Au particle density.

The nanostructure changes significantly when the film is deposited using a deposition rate of 5 Hz. The TEM/STEM and EDX images in Figure 3 show the Au and Fe phases appear to have become well-mixed as their composition maps completely overlap based on both the cross-sectional and plan-view EDX mapping (Figure 3C–F,J–M). The nanoparticles composed of Au and Fe at 5 Hz have an average diameter of 18.7 nm as measured from the plan-view images. The nanoparticles are more elongated rather than spherical as seen in the cross-section images in Figure 3A–F. The 3D schematic in Figure 3G also reflects this. The amount of Au in the nanoparticles has increased with increased deposition rate (Figure 3J–M). The increase in the Au content could be due to a change in the pulse distribution between the Au strip and the BTO-Fe composite target at the different frequencies. The reflectivity of the Au strip itself could also reduce the amount of Au deposited. The distribution of the bimetallic nanostructures across the thin films is also well-distributed across the thin film. The resultant nanostructure of the sample deposited at 10 Hz is quite similar to the nanostructure of the 5 Hz sample, where the Au and Fe phases are well-mixed, as shown in the plan-view and cross-section views of the TEM/STEM/EDX analysis (Appendix A). The nanoparticles composed of Au and Fe grown at 10 Hz have an average diameter of 11.1 nm measured from the plan-view images (Appendix A). This average size change follows the general trend of nanoscale features shrinking with reduced diffusion time, as the laser frequency increases. The Au-Fe alloyed nanoparticles grown at 10 Hz are more spherical than those grown at 5 Hz (Appendix A). The plan-view images also show similarly even distribution across the thin film of the Au-Fe alloyed nanostructures. This suggests that higher deposition rate did not allow for the Au and Fe phases to segregate into their thermodynamically preferred immiscible phases; instead, a well-mixed Au-Fe phase has formed in both the 5 Hz and 10 Hz samples.

The X-ray diffraction (XRD) *θ*-2*θ* pattern (Figure 4) shows some shift in the Au and Fe peaks to higher 2*θ* values as well as peak broadening as the deposition rate increases. Increasing 2*θ* values and broader peaks indicate that some of the metal lattices could be under compressive out-of-plane strain due to both metallic phases being vertically coupled in the BTO matrix as seen in Figure 2 and Appendix A. The full XRD spectrum is included in Appendix A. Previous work with Au-Fe systems grown under non-equilibrium conditions has seen a similar peak shift to the right and peak broadening that suggests the formation of an Au-Fe alloy [11,33,37]. When the ratio of the Au:Fe is less than 50%, there is a peak splitting between the Au peak and the alloy peak [32,37]. The increased frequency during deposition increased the percentage of Fe present in the thin film materials, therefore increasing the compressive strain of the metals were under and decreasing the overall lattice parameter. The peak split in the 10 Hz grown sample also indicates behavior similar to other non-equilibrium growth techniques implementing lasers and chemical synthesis to create non-equilibrium bimetallic nanoparticles [11,13,32,38]. The Fe peak intensity in the 2 Hz sample has a significantly lower intensity due to being at the bottom of the film, underneath the layer of Au particles and BTO matrix. The BTO (002) peaks also show slight changes indicating that the crystallinity of the BTO matrix could be affected by the metal nanostructures embedded within.

To explore the impacts of the different phases and morphologies of the Au-Fe pillars on properties, each sample was measured for their magnetic hysteresis loop as shown in Figure 5. For the sample grown at the lowest deposition rate (2 Hz) with obvious phase separation, the magnetic saturation and coercivity were the most consistent in both the in-plane (IP) and out-of-plane (OP) directions at both 10 K and 300 K, i.e., isotropic magnetic properties. Specifically, for the 2 Hz sample, the IP magnetic saturation is 19.5 emu/cm^3^ and 17.4 emu/cm^3^ at 10 K and 300 K, respectively. The IP coercivity is 390 Oe and 155 Oe at 10 K and 300 K, respectively. The OP magnetic saturation is 19.5 emu/cm^3^ and 17.1 emu/cm^3^ at 10 K and 300 K, respectively. The OP coercivity is 310 Oe and 93 Oe at 10 K and 300 K, respectively. For this work in growing an Au-Fe-BTO system, the higher deposition rates produce thin film materials with much larger differences in their magnetic behavior in terms of their IP and OP directions, i.e., magnetic anisotropy. For the 5 Hz sample, the IP magnetic saturation is 38.6 emu/cm^3^ and 11.4 emu/cm^3^ at 10 K and 300 K, respectively. The IP coercivity is 138.5 Oe and 37.9 Oe at 10 K and 300 K, respectively. The OP magnetic saturation is 45.7 emu/cm^3^ and 12.8 emu/cm^3^ at 10 K and 300 K, respectively. The OP coercivity is 300.9 Oe and 16.9 Oe at 10 K and 300 K, respectively. Interestingly, the sample grown at 10 Hz shows enhanced anisotropic behavior with significantly lower magnetic response in the IP direction. For the 10 Hz sample, there is a larger difference in magnetic behavior in the OP direction at the different temperatures. The IP magnetic saturation is 1.7 emu/cm^3^ and 1.7 emu/cm^3^ at 10 K and 300 K, respectively. The IP coercivity is 160 Oe and 155 Oe at 10 K and 300 K, respectively. The OP magnetic saturation is 53.3 emu/cm^3^ and 13.8 emu/cm^3^ at 10 K and 300 K, respectively. The OP coercivity is 190 Oe and 38 Oe at 10 K and 300 K, respectively. Overall, the magnetic response of the completely separated particles in the 2 Hz sample was the least anisotropic and had the largest response at room temperature. This is due to the spherical-shaped Fe nanoparticles in the nanocomposite. As the deposition rate increased, the anisotropy increased. The OP magnetic saturation also increased with deposition rate at low temperature. The increased amount of Au and alloying caused this change in anisotropic magnetic behavior as the shape of the alloyed nanoparticles become more rod-like, which presents more shape anisotropy.

Au is a well-known plasmonic material, so measuring the change in the optical properties with various bimetallic nanostructures is of interest in addition to their magnetic properties. Figure 6 shows the optical permittivity of each film in both in-plane (i.e., parallel) and out-of-plane (i.e., perpendicular) components. For all three films, the parallel component is positive in each film and the perpendicular component becomes negative at different wavelengths, which suggest anisotropic optical permittivity in certain optical wavelength ranges, i.e., hyperbolic properties. This is resulted from the anisotropic structures of metallic nanostructures in dielectric BTO matrix. For the 2 Hz film with Au nanoparticles on top, the film shows hyperbolic behavior in the wavelength range beyond 1732 nm. A similar behavior is seen across the other films with different epsilon near zero (ENZ) points of 965 nm and 1552 nm for the alloyed 5 Hz and 10 Hz bimetallic nanostructures, respectively. The sample grown at 5 Hz with the alloyed nanostructures that were the largest in average size shows the broad wavelength range of hyperbolic behavior from 965 nm and beyond, except a very small isotropic range of 1594–1684 nm. The broad hyperbolic behavior in the 5 Hz sample is due to the larger amount of Au present in the alloy and more vertically aligned Au-Fe nanostructures in the 5 Hz sample than those of the 2 Hz and 10 Hz samples. The more elongated, larger alloyed Au-Fe nanoparticles of the 5 Hz sample compared to the 10 Hz sample create the additional hyperbolic transition point in the permittivity data. Although the alloyed nanostructures appear to have shifted the hyperbolic wavelength down, each film only behaves hyperbolically in the infrared region. Appendix A shows the transmittance plot for each film, where the transmittance values vary drastically due to the very different morphology and distribution of the Fe-Au bimetallic nanoparticles in each sample.

Overall, it is interesting to observe obvious phase segregation property changes in Au-Fe-BTO three-phase nanocomposite systems simply by varying the laser deposition rate. The 2 Hz sample results in separate Au and Fe nanoparticles in the matrix and the least magnetic and optical anisotropy. While the 5 Hz and 10 Hz samples result in much more uniformly alloyed Au-Fe nanostructures in the BTO matrix and stronger magnetic and optical anisotropy. In previous work with the three-phase nanocomposite Au-Fe-LSFO, the system was grown with an LSFO target (1:1 ratio of LaSr:Fe_2_O_3_) with an Au strip on top. The Fe self-segregated out of an LSFO target, unlike the composite target used in this work [24]. In the work with the nanocomposite thin film composed of an Au-Ag bimetallic alloy in a ZnO matrix, the Ag and Au precursors were separate from the ZnO [27]. The Fe separating itself from an oxide source could contribute to the metal not being able to alloy uniformly under non-equilibrium conditions, as shown in this work. The additional energy input into the growth process puts enough energy in the system for the Au-Fe-BTO and Ag-Au-ZnO systems to create bimetallic alloys due to the metallic phases already being separate. The additional energy added from PLD, however, is not enough for the Fe to separate from the Fe_2_O_3_ and alloy with the Au. Alloyed Au-Fe nanostructures are possible under the fast deposition followed by the rapid solidification process during PLD, without the intermediate step of Fe decomposition process involved in the LSFO-Fe-Au case. Besides Au and Fe, many other immiscible systems are worth exploring. For example, Au-Co, Ag-Co, Cu-Co, and Cu-Fe systems, are all plasmonic-magnetic metal systems worth exploring in an oxide matrix. These new alloyed bimetallic systems could be of great interest in exploring tunable magnetic, optical, and magneto-optical coupling properties that could not be achieved easily with their immiscible nature under equilibrium conditions. Because of their uniquely coupled magnetic-plasmonic properties, these non-equilibrium bimetallic nanostructures could find various applications such as energy efficient optical switchable spintronics, logic devices, and sensors.

## 3. Conclusions

Non-equilibrium fabrication techniques produce unique phases that cannot be achieved under standard equilibrium methods. As a demonstration, PLD was used to create well-mixed alloyed Au-Fe nanoparticles in a BTO matrix through increasing the deposition rate from 2 Hz to 10 Hz and limiting the diffusion time of the adatoms. Alloying the nanoparticles increased their magnetic anisotropy (IP vs. OP) and low-temperature magnetic saturation. The influence of the alloy on the optical permittivity properties lowered the hyperbolic transition wavelength from 1732 nm in 2 Hz sample to 1552 nm in 10 Hz sample. Based on the alloyed bimetallic nanoparticles of Au-Fe, three-phase designs by PLD could be a viable non-equilibrium method to alloy other immiscible metals into a variety of alloyed nanostructures for novel physical properties that could not be achieved in equilibrium systems.

## 4. Experimental Methods

Pulsed laser deposition (PLD) with a KrF laser (Lambda Physik, Göttingen, Niedersachsen, Germany, *λ* = 248 nm) at varying frequency of 2, 5, and 10 Hz was used to fabricate the thin film samples. The laser energy was set at 420 mJ. The power density was set at 1.5 J/cm^2^. The nanocomposite thin films were made with a composite target of BTO:Fe with molar ratio of 1:1 and an Au metal strip on top, grown on single-crystal SrTiO_3_ (STO) (001) substrates under vacuum at 750 °C. The chamber was pumped down until it achieved a pressure of ~10^−7^ Torr before heating the chamber to 750 °C. After deposition, the thin films were cooled at 20 °C/min. The BTO-Fe composite target was made through mixing BTO and Fe powders with a 1:1 molar ratio, pressing the powder into a disk, and sintering in a spark plasma sintering tool. The Au metal strip was attached to the BTO-Fe target with Ag paste and polyimide tape.

X-ray diffraction (XRD) (PANalytical Empyrean diffractometer, Malvern, UK) was used to determine the crystallography of the thin film samples using *θ*–2*θ* scans measuring from 20–80° and comparing the results to existing pdf cards. Transmission Electron Microscopy (TEM) (FEI TALOS F200X, Waltham, MA, USA) was used to determine the differences in nanostructure and phase distribution of the films with energy-dispersive X-ray spectroscopy (EDX) mapping and high-resolution scanning transmission electron microscopy (STEM). TEM samples were created in both cross-section and plan-view orientations with the standard manual preparation process that includes grinding, dimpling, and ion milling (PIPS II Gatan).

A Magnetic Property Measurement System (MPMS) (MPMS Model 3, Quantum Design) measured the magnetic properties for in-plane and out-of-plane directions of each film. The magnetic hysteresis was measured at room-temperature and 10 K from 10,000 Oe to −10,000 Oe. Optical property measurements were conducted using a RC2 spectroscopic ellipsometer (J.A. Woollam Company, Lincoln, NE, USA) where the angles measured were 30, 45, and 60° with a spectrum range of 210–2500 nm. The psi and delta data were fit with a B-spline model to determine anisotropic permittivity.

## Figures and Tables

**Figure 1 nanomaterials-12-03460-f001:**
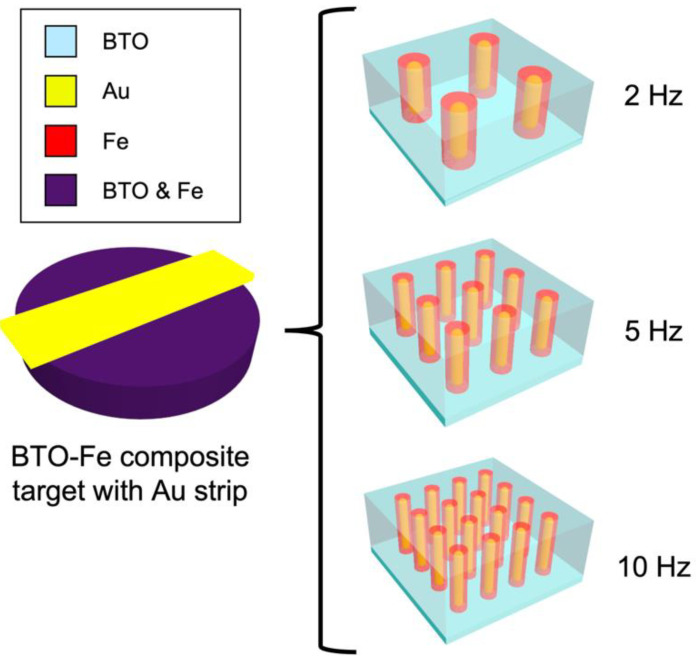
Predicted film fabrication at each deposition frequency based on using a single BTO-Fe target with an Au strip.

**Figure 2 nanomaterials-12-03460-f002:**
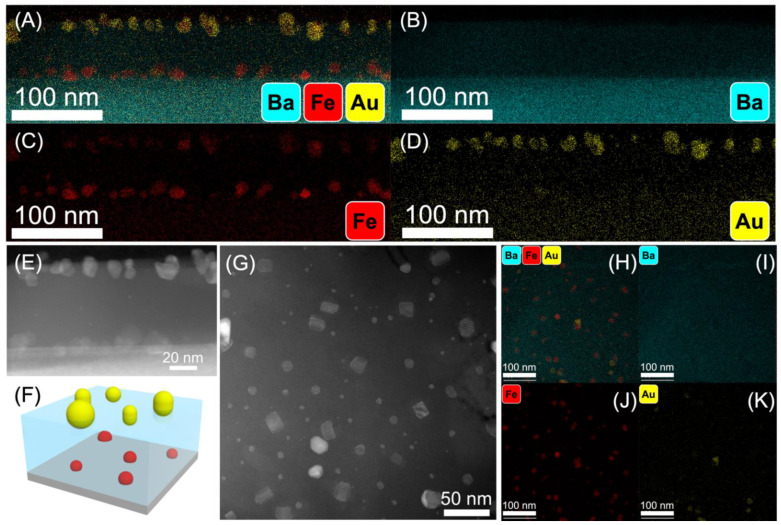
TEM results of the 2 Hz sample: (**A**) cross-section EDX composite of Ba, Fe, and Au, (**B**) cross-section EDX of Ba, (**C**) cross-section EDX of Fe, (**D**) cross-section EDX of Au, (**E**) cross-section STEM image, (**F**) 3D schematic of 2 Hz sample, (**G**) plan-view STEM image, (**H**) plan-view EDX composite of Ba, Fe, and Au, (**I**), EDX composite of Ba, (**J**) EDX composite of Fe, (**K**) and EDX composite of Au.

**Figure 3 nanomaterials-12-03460-f003:**
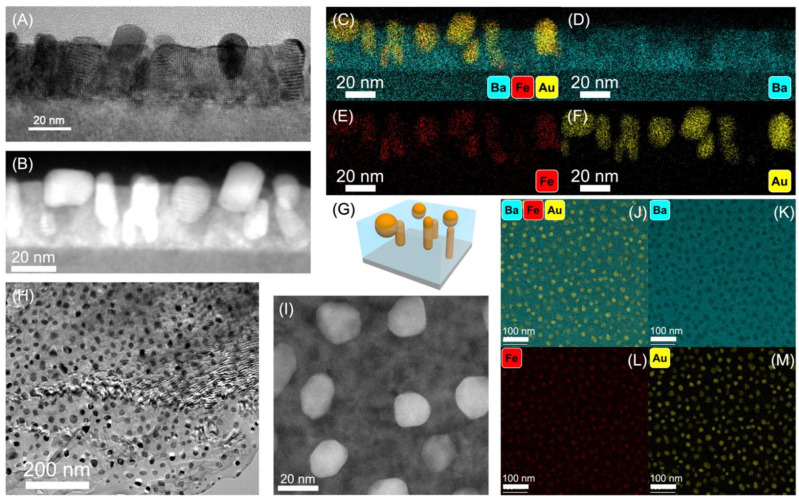
TEM results of 5 Hz sample: (**A**) cross-section TEM image, (**B**) cross-section STEM image, (**C**) cross-section EDX composite of Ba, Fe, and Au, (**D**) cross-section EDX of Ba, (**E**) cross-section EDX of Fe, (**F**) cross-section EDX of Au, (**G**) 3D schematic of 5 Hz sample, (**H**) plan-view TEM image, (**I**) plan-view STEM image, (**J**) plan-view EDX composite of Ba, Fe, and Au, (**K**), EDX composite of Ba, (**L**) EDX composite of Fe, (**M**) and EDX composite of Au.

**Figure 4 nanomaterials-12-03460-f004:**
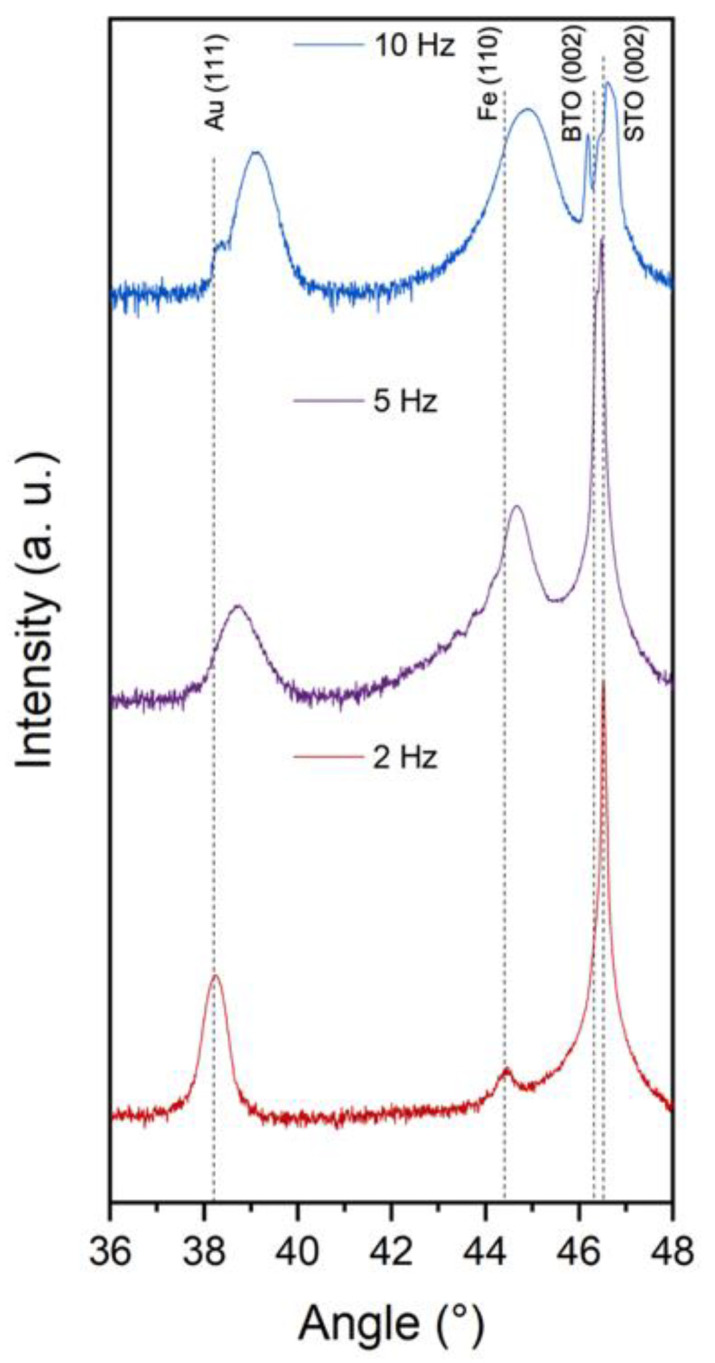
XRD *θ*–2*θ* results for all three samples deposited under different laser frequency.

**Figure 5 nanomaterials-12-03460-f005:**
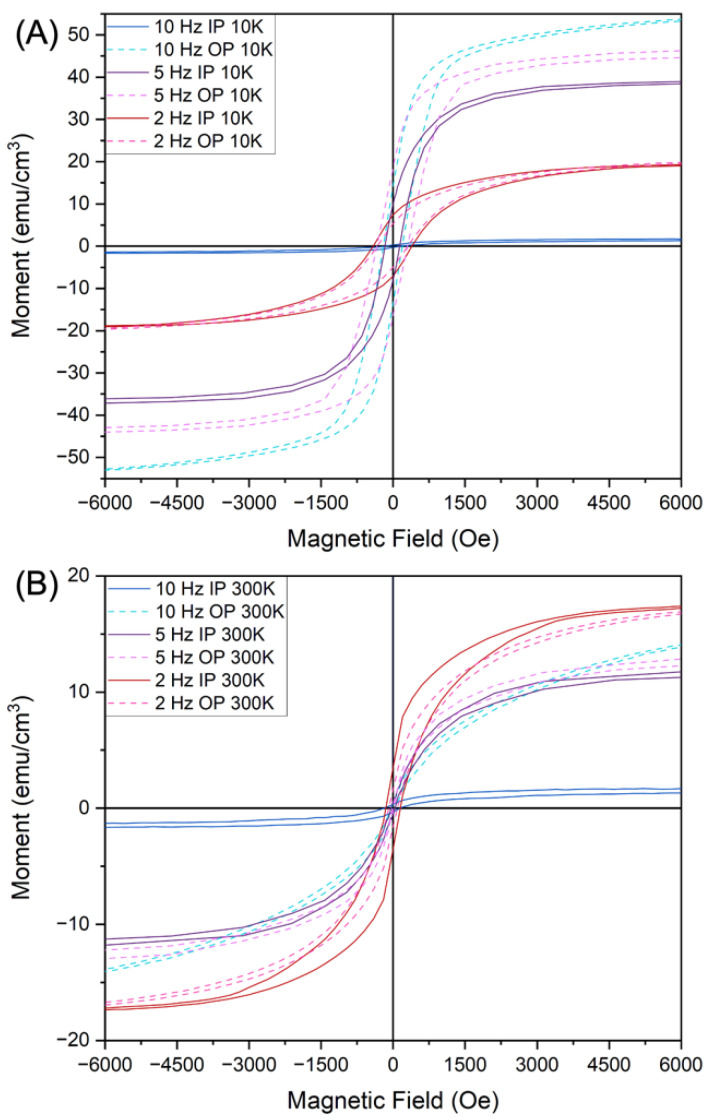
Magnetic hysteresis loops measured IP and OP of each sample (**A**) at 10 K and (**B**) at 300 K.

**Figure 6 nanomaterials-12-03460-f006:**
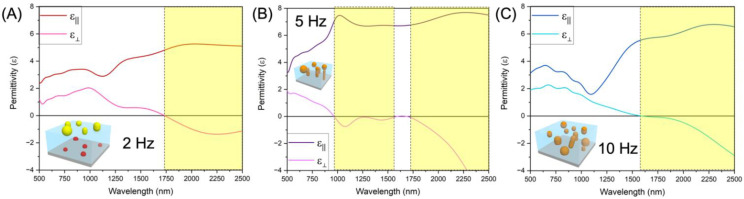
Optical permittivity of the Au-Fe-BTO sample grown at (**A**) 2 Hz, (**B**) 5 Hz, (**C**) and 10 Hz.

## Data Availability

Not applicable.

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
