# Peer review of "Single-Step Fabrication of Au-Fe-BaTiO3 Nanocomposite Thin Films Embedded with Non-Equilibrium Au-Fe Alloyed Nanostructures"

_nanomaterials, 2022, doi:10.3390/nano12193460_

Round 1

Reviewer 2 Report

The manuscript entitled "Single-step fabrication of Au-Fe-BaTiO3 nanocomposite thin films embedded with non-equilibrium Au-Fe alloyed nanostructures" contains the results of a study devoted to the selection of conditions and analysis of the results of the synthesis of new functionalized composite materials consisting of components immiscible in equilibrium systems. This work is of scientific interest because it illustrates a number of important principles using modern methods. It makes sense to add references to literary sources. The text is clear, the narrative is logical. It is proposed to accept the article after minor revision.

1) Lines 90, 158, 189 – too much free space at the end of the page, not occupied by the text.

2) Line 36. We recommend that you add to the list of works an article devoted to the study of a composite material made of magnetic nanoparticles and DNA with a nanotopography of the surface. https://doi.org/10.3390/polym14020344

3) It is not clear how the schematic picture in Figure 1 is related to the authors' research. In other illustrations, the composite structure differs from the predicted one.

4) Mention should be made of the possible limitations of the method. In the controlled synthesis of such functionalized composite materials, is there a maximum layer thickness? And if, according to the authors, there are restrictions, then what are they?

5) To what extent can the disequilibrium of systems be linked to the regularity of the structure of the material, which the authors stated in the introduction? Metastable equilibrium is the opposite of constancy. In Figures 2 and 3, you can see just irregular structures.

Reviewer 3 Report

The authors presented Single-step fabrication of Au-Fe-BaTiO3 nanocomposite thin films embedded with non-equilibrium Au-Fe alloyed nanostructures. I have read and checked the manuscript. Some minor revisions are provided. Please check the manuscript thoroughly & make the necessary correction.

1. Authors need to use passive sentences. Avoid using 'we,' 'us' etc. 

2. Language shall be better checked.

3. Check that all typographical errors are removed from the text. 

4. Although the study comprises a considerable number of references. Still, some latest references are missing. It can be improved. 

5. The experimental techniques need to be explained further. 

6. Discussion; the scales for all physical quantities must be presented in figures; the quality of the figures is good.   
